



**Effect of ocean acidification on the structure and fatty acid composition**
**of a natural plankton community in the Baltic Sea**
J.R. Bermúdez[1,2], M. Winder[3], A. Stuhr[1], A.K. Almén[4,5,6], J. Engström-Öst[4,5], U. Riebesell[1]
[1]GEOMAR Helmholtz Centre for Ocean Research Kiel, Germany.
[2] {Facultad de Ingeniería Marítima, Ciencias Biológicas, Oceánicas y Recursos Naturales.
Escuela Superior Politécnica del Litoral, ESPOL, Guayaquil, Ecuador}
[3] {Department of Ecology, Environment and Plant Sciences, Stockholm University,
Stockholm, Sweden}
[4] {Environmental and Marine Biology, Faculty of Science and Engineering, Åbo Akademi University,
Åbo, Finland}
[5] {Aronia Research and Development Institute, Novia University of Applied Sciences and Åbo
Akademi University, Ekenäs Finland}
[6] {Tvärminne Zoological Station, University of Helsinki, J.A. Palménin tie 260, FI-10900 Hanko,
Finland}
Correspondence to: J.R. Bermúdez (jrbermud@espol.edu.ec)

**Keywords**
Fatty acids, *Acartia bifilosa*, *Eurytemora affinis*, plankton community, $CO_2$, ocean
acidification, Baltic Sea.

**Abstract**
Increasing atmospheric carbon dioxide ($CO_2$) is changing seawater chemistry towards reduced
pH, which consequently affects various properties of marine organisms. Coastal and brackish
water communities are expected to be less affected by ocean acidification (OA) as these
communities are typically adapted to high fluctuations in $CO_2$ and pH. Here we investigate the
response of a coastal brackish water plankton community to increasing $CO_2$ levels as projected
for the coming decades and the end of this century in terms of community and biochemical fatty
acid (FA) composition. A Baltic Sea plankton community was enclosed in a set of off-shore
mesocosms and subjected to a $CO_2$ gradient ranging from natural concentrations (~347 µatm
$p$CO$_2$) up to values projected for the year 2100 (~1333 µatm $p$CO$_2$). We show that the
phytoplankton community composition was resilient to $CO_2$ and did not diverge between the
treatments. Seston FA composition was influenced by community composition, which in turn



was driven by silicate and phosphate limitation in the mesocosms, and showed no difference
between the $CO_2$ treatments. These results suggest that $CO_2$ effects are dampened in coastal
communities that already experience high natural fluctuations in $pCO_2$. Although this coastal
plankton community was tolerant to high $pCO_2$ levels, hypoxia and $CO_2$ uptake by the sea can
aggravate acidification and may lead to pH changes outside the currently experienced range for
coastal organisms.

**1 Introduction**

The steady increase of carbon dioxide ($CO_2$) due to anthropogenic emission since the beginning
of the industrial era has increase the atmospheric concentration (Boyd et al. 2014). The ocean
has a large carbon sink capacity, and increasing atmospheric $CO_2$ absorbed by the ocean is
changing the chemistry of the seawater, causing a decline in pH termed Ocean Acidification
(OA) (Boyd et al. 2014). OA has been shown to affect various biological processes of diverse
marine species (Doney et al. 2009; Kroeker et al. 2010). For instance OA can impact the
biochemical and elemental composition of organisms (Sato et al. 2003; Torstensson et al. 2013),
which can be transferred to higher trophic levels (Rossoll et al. 2012). OA can also drive
alterations in the community composition structure of primary producers (Hare et al. 2007;
Biswas et al. 2011; Schulz et al. 2013). Strong $CO_2$-effects may be particularly significant in
marine species that experience low natural fluctuations in $CO_2$ (Riebesell et al., in review). In
contrast, coastal and brackish-water environments encounter wide and frequent fluctuations in
$pCO_2$ (Hinga 2002; Rossoll et al. 2013), due to large fluxes of organic and inorganic carbon
from river runoff and lower alkalinity, and hence reduced buffer capacity (Melzner et al. 2013).
Consequently, it can be expected that coastal and brackish communities are more tolerant to
OA effects (Rossoll et al. 2013; Reusch & Boyd 2013) and adverse $CO_2$ effects in terms of the
biochemical composition of primary producers and variations in community composition may
be diminished.

Fatty acids (FA) are the main components of lipids in cell membranes. In particular
polyunsaturated fatty acids (PUFA) have important physiological roles in algae, which
synthesize them in high amounts. Heterotrophs at higher trophic levels cannot synthesize
certain FA *de novo*, especially PUFA, and have to acquire them from dietary sources (Izquierdo
et al. 2001). Diverse laboratory studies of monocultures showed that $CO_2$ alters the FA profile
of individual algal species (Sato et al. 2003; Fiorini et al. 2010; Torstensson et al. 2013;





Bermúdez et al. 2015). A $CO_2$-driven change in algal food quality can be detrimental for
grazers, as has been shown in a laboratory study under elevated $CO_2$ levels (Rossoll et al. 2012).
A strong decline of PUFA in a diatom, grown at high $CO_2$ affected the FA composition of
copepods grazing on them and severely impaired their development and egg production rates.
Furthermore, increasing seawater $CO_2$ can modify phytoplankton community composition by
favoring certain taxa of primary producers (Graeme et al. 2005). In particular, small-sized cells
benefit from high $CO_2$ (Hare et al. 2007; Biswas et al. 2011; Brussaard et al. 2013). This is
ecologically relevant as taxonomic phytoplankton groups have contrasting FA profiles
(Galloway & Winder 2015) and a change in community structure can affect higher trophic
levels. For instance, a field study of two cladocerans having different phytoplankton
composition as food source showed decreased egg production, lipid reserves, body size and
abundance when fed with algae from an acidic lake (Locke & Sprules 2000).

The above observations suggest that changes in planktonic biochemical makeup and associated
shifts in community composition of primary producers as a result of OA can affect the transfer
of essential compounds to upper trophic levels. However, organisms and communities from
coastal/brackish environments may show a high tolerance to elevated $p$CO$_2$ levels due to
adaptation (Thomsen et al. 2010; Nielsen et al. 2010; Rossoll et al. 2013). In coastal/brackish
systems variation in $CO_2$ is more frequent and severe due to river runoff (Hinga 2002), reduced
buffer capacity (Feely et al. 2004), seasonal processes (Melzner et al. 2013)and upwelling of
$CO_2$ enriched water (Feely et al. 2009), all of which lead to wider pH variation in coastal
systems compared to the open ocean (Hinga 2002). Laboratory studies have shown that algae
subjected to long-term high $CO_2$ levels can restore their physiological optima through adaptive
evolution (Lohbeck et al. 2012; Bermúdez et al. 2015) and that coastal communities are resilient
to OA-driven changes in community composition and biomass (Nielsen et al. 2010; Rossoll et
al. 2013). Therefore, it can be expected that organisms in these areas are adapted to high $CO_2$
fluctuations, hampering any $CO_2$-driven effects previously observed in plankton communities
(Locke & Sprules 2000; Biswas et al. 2011).

The goal of the present study was to determine if an increase in $CO_2$ affects phytoplankton
community composition and their FA composition, and if any effects are transferred to grazers
of a natural plankton community in a coastal/brackish environment. A set of off-shore
mesocosms, that enclosed a natural plankton assemblage of the Baltic Sea, were used as
experimental units. The $CO_2$ levels ranged from current to projected next century values (Boyd





et al. 2014, scenario A2). Algal FA were measured in total seston and in the copepods *Acartia*
*bifilosa* and *Eurytemora affinis*, respectively, which are dominant zooplankton in this
ecosystem (Almén et al. 2015).

**2 Material and Methods**

**2.1 Experimental set-up and CO$_2$ manipulation**
Our study was conducted during an off-shore CO$_2$ mesocosm perturbation experiment off the
Tvärminne Zoological Station at the entrance to the Gulf of Finland at 59° 51.5' N, 23° 15.5'
E during late spring 2012. We used six enclosures with a length of 17 m containing ~55 m$^3$ of
natural sea water (Paul et al. 2015). The mesocosms were set up and manipulated as described
in detail by Paul et al. (2015) and Riebesell et al. (2013). Carbon dioxide enrichment was
achieved in two phases through the addition of CO$_2$-saturated seawater to four out of six
mesocosms. In phase 1, CO$_2$ was added in five steps between day 1 and day 5 to achieve values
from ambient (~240 µatm) and up to ~1650 µatm $f$CO$_2$. In phase 2 at day 15 CO$_2$ was again
added in the upper 7 m to compensate for pronounced outgassing in the CO$_2$ enriched
mesocosms. Samples for phytoplankton counts were taken every second day and for fatty acid
concentrations every fourth day using a depth-integrated water sampler (Hydrobios, Kiel,
Germany) covering the upper 15 m of the water column. Integrated zooplankton net tows were
taken every seventh day.

**2.2 Phytoplankton abundance and biomass calculation**
Phytoplankton cell counts up to a cell size of 200 µm were carried out from 50 ml water
samples, fixed with alkaline Lugol's iodine (1% final concentration) using the Utermöhl's
(1958) method with an inverted microscope (ZEISS Axiovert 100). At 200 times magnification,
cells larger than 12 µm were counted on half of the chamber area, while smaller cells were
counted at 400 times magnification on two radial strips. The plankton was identified to genus
or species level according to Tomas (1997), Hoppenrath et al. (2009) and Kraberg et al. (2010).
Algal biovolume was calculated according to geometric shapes and converted to cellular
organic carbon using taxon-specific conversion equations for phytoplankton (Menden-Deuer &
Lessard 2000).





### 2.3 Fatty acid composition

For analysis of algal fatty acid (FA), 500 ml of seawater were filtered by using pre-combusted (450°C, 6 h) Whatman GF/F (~0.7 μm pore size) filters. For zooplankton gravid copepod females of *Acartia bifilosa* and *Eurytemora affinis* were picked with sterile tweezers under two stereomicroscopes (Nikon SMZ800, 25× magnification and Leica 25× magnification) and placed in pre-weighted tin cups. All samples were immediately stored at -80°C until analysis. FA were measured by gas chromatography as fatty acid methyl esters (FAMEs) following Breteler et al. (1999). Lipids were extracted over night from the filters using 3 ml of a solvent mixture (dichloromethane:methanol:chloroform in 1:1:1 volume ratios). As internal standard, FAME C19:0 (Restek, Bad Homburg, Germany; c= 20.0 ng component$^{-1}$μl$^{-1}$) was added, and a C23:0 FA standard (c= 25.1 ng μl$^{-1}$) used as an esterification efficiency control (usually 80-85 %). Water-soluble fractions were removed by washing with 2.25 ml of KCl solution (c= 1 mol L$^{-1}$), and the remainder dried by addition of NaSO$_4$. The solvent was evaporated to dryness in a rotary film evaporator (100-150 mbar), re-dissolved in chloroform and transferred into a glass cocoon. The solvent was evaporated again (10-30 mbar), and esterification was performed overnight using 200 μl 1% H$_2$SO$_4$ (in CH$_3$OH) and 100 μl toluene at 50°C. Phases were split using 300 μl 5% sodium chloride solution, and FAMEs were separated using n-Hexane, transferred into a new cocoon, evaporated, and 100 μl (final volume) added. All solvents used were gas chromatography (GC) grade. FAME were analyzed by a Thermo GC Ultra gas chromatograph equipped with a non-polar column (RXI1-SIL-MS 0.32 μm, 30 m, company Restek) and Flame ionization detector. The column oven was initially set to 100°C, and heated to 220 °C at 2 °C min$^{-1}$. The carrier gas was helium at a constant flow of 2 ml min$^{-1}$. The flame ionization detector was set to 280 °C, with a gas flow of 350, 35 and 30 ml min$^{-1}$ of synthetic air, hydrogen and helium, respectively. A 1 μl aliquot of the sample was injected. The system was calibrated with a 37-component FAME-mix (Supelco, Germany) and chromatograms were analyzed using Chrom-Card Trace-Focus GC software (Breteler & Schouten, 1999) and the fatty acids were clustered according to their degree of saturation: saturated (SFA), monounsaturated (MUFA) and polyunsaturated (PUFA).

### 2.4 Statistical analyses

The data was analyzed by a nested Mixed Effects ANOVA Model (LME) to determine the differences in taxa biomass (μgC ml$^{-1}$) and relative fatty acid content (% in the seston and zooplankton) between the CO$_2$ treatments (μatm $f$CO$_2$), with $f$CO2, silicate, inorganic nitrogen (nitrite + nitrate), phosphate, temperature and salinity as fixed effects, and sampling day and



mesocosm position as nested random variable (random distribution of $CO_2$ treatments among
the mesocosm). Average mesocosm $f$CO$_2$ was calculated for the total duration of the sampling
period plankton community composition (day 1 to 29) and for FA data analysis (day 1 to 25 for
seston FA and day -1 to 33 for zooplankton FA). Linear regression models were used to
determine the relation between PUFA and phytoplankton biomass. The similarity in the
structure of the plankton community between the treatments was analyzed by Non Metrical
Multidimensional Scaling (NMDS) with Bray distance, auto-transformation and 3 dimensions
(k=3). This analysis distributes the samples in an ordination space according to the biomass of
the different taxa in the community along orthogonal principal components using non Euclidean
distances for ordination space, which makes it more robust to the presence of zero values
(Clarke 1993). All statistical analyses were done using the R software environment 3.0.1 (R
Development Core Team 2013).

**3 Results**

**3.1 Plankton community composition**
The initial algal community consisted of post-bloom species dominated by small-sized cells,
with dinophyta being the most abundant phytoplankton group in all mesocosm throughout the
experiment followed by heterokontophyta, euglenophyta, cholorophyta, cyanobacteria bigger
than 5µm (usually filamentous) and small abundances of cryptophya (Fig. 1).
Microzooplankton was present during the entire experimental period, particularly the
choanoflagellate *Calliacantha natans* (Fig. 1). The plankton community was analyzed from day
1 to 29, which comprised two phases as described by Paul et al. (2015), with a Phase 1 (from
day 1 to 15) where phytoplankton biomass gradually increased until day 10 when a bloom
started and reached a peak around day 15 in all treatments; while in a Phase 2 (from day 17 to
29) the biomass began to decay from around day 19 up to day 29 (Fig. 1).

The more abundant taxa did not show differences in abundance between the $CO_2$ treatments on
both phases (Fig. 2a, b). However, the biomass of some of the less abundant groups was affected
by $CO_2$ within the different phases. In Phase 1, the nested mixed effects model analysis of the
algal biomass showed that chlorophyta decrease significantly towards high $CO_2$ levels (Fig. 2a)
(LME, F= 7.27, p= 0.01, df= 20). Nevertheless, there was a difference in the relative biomass
of the more abundant plankton groups between Phases 1 and 2, with a decrease in dinophyta
(37.2 ± 3.2 % to 28.3 ± 2.9 %) and heterokontophyta (19.1 ± 2.2 % to 14 ± 2.6) (Fig. 2c) and



an increase of euglenophyta (7.5 ± 1.4 % to 21 ± 2.7) and chlorophyta (14.0 ± 1.5 % to 19.1 ±
2.4) (Fig. 2d). An NMDS analysis of the entire phytoplankton community showed a rather
homogeneous community composition between the different $CO_2$ treatments but variation
among sampling days, especially at day 7, when the diatom *Melosira varians* was abundant
during that particular sampling day (Fig. S1).

**3.2 Seston fatty acid composition**

The PUFA represented on average ~26 ± 4%, MUFA ~22 ± 3% and SFA ~52 ± 4% of the total
FA content in the seston over the entire experimental period. The Mixed Effect Model (LME)
analysis of relative PUFA content data showed no significant difference among the $CO_2$
treatments (LME, $F_{45}$= 0.0, p>0.05) (Fig. 3a PUFA). The MUFA and SFA did neither show any
significant change in abundance in relation with $CO_2$ (LME, $F_{45}$= 0.0, p= 0.8, and $F_{45}$= 0.06, p=
0.79, respectively) (Fig. 3a MUFA, SFA). However, the FA composition of the seston showed
that the relative PUFA content significantly decreased over time in all mesocosms (linear
regression, $R^2$= 0.52, t= -7.64, p<0.0001, n=22) (Fig. 3b High $CO_2$ treatments, Low $CO_2$
treatments), while the MUFA and SFA increased, although the relation of both with time is
weak (linear regression, $R^2$= 0.12, t= 2.88, p= 0.005 and $R^2$= 0.15, t= 3.26, p= 0.001, n=22
respectively) (Fig. S2).

Nevertheless, PUFA showed a positive relation with heterokontophyta (linear regression,
$R^2$=0.58, p<0.001) and dinophyta (linear regression, $R^2$=0.41, p<0.001) biomass (Fig. 4a); and
with silicate (LME, F= 22.8, p< 0.001, df= 35) and phosphate (LME, F= 9.3, p< 0.01, df= 35)
abundance in the mesocosms (Fig. 4b).

**3.3 Copepod fatty acids**

The overall PUFA content of the copepod *A. bifilosa* represented ~12% (311 ± 175 ng FA mg
dry wt.$^{-1}$) and in *E. affinis* ~16% (433 ± 597 ng FA mg dry wt.$^{-1}$) of the total FA.

The FA did not show a $CO_2$-related effect in *A. bifilosa* (LME, F= 0.62, p= 0.4374, df= 26)
(Fig. 5a), or *E. affinis* (F= 0.13, p= 0.71, df= 26) (Fig. 5b). Nevertheless the relative PUFA
content in *A. bifilosa* and *E. affinis* showed a significant decrease over time in all high and low
$CO_2$ treatments (linear regression, *A. bifilosa*; $R^2$= 0.22, t= -3.288, p= 0.002 *E. affinis;* $R^2$= 0.47,
t= -5.51, p< 0.0001 ) (Fig. 5c), while MUFA and SFA increased in both species (Fig. S3).
Furthermore, the relative FA content in *E. affinis* varied over time following the changes in the



seston FA, this relation was significant but weak for PUFA MUFA and SFA (Fig. S4), while in
*A. bifilosa* this change appeared only in the MUFA (Fig. S4).

**4. Discussion**

**4.1 Community composition**
The plankton community composition in the present experiment did change over time and
showed little differences in relation to the different $CO_2$ treatments. The observed absence of a
strong $CO_2$ effect on the community composition in the present study is in line with the
observations in the western Baltic Sea (Thomsen et al., 2010; Nielsen et al., 2010; Rossoll et
al., 2013). In these studies the authors suggested that the plankton community is adapted to OA
due to the recurrent large seasonal and daily variance of pH and $CO_2$ experienced by the
communities in this productive low-salinity region (Thomsen et al. 2010; Nielsen et al. 2010;
Rossoll et al. 2013; Almén et al. 2014). Our study region, a coastal zone in the western Gulf of
Finland in the northern Baltic Sea, is a brackish environment with low salinity (~5.7 ‰), a high
fresh water runoff (~111 $km^3$ $year^{-1}$) (Savchuk 2005) and a strong inter- and intra-seasonal pH
variability, sometimes reaching extreme values of 9.2 and 7.4 with an average of 8.1 (Brutemark
et al. 2011). Therefore, it seems that the plankton community in our study area, which
experiences high natural pH fluctuations, is composed of species and genotypes that are less
pH/$CO_2$ sensitive (Nielsen et al. 2010; Lohbeck et al. 2012; Melzner et al. 2013; Rossoll et al.
2013) which allows them to cope with the $CO_2$ range applied in the current field experiment.

Chlorophytes were the only group that showed a significant response to the $CO_2$ treatment,
although their contribution to total biomass was low. Chlorophytes decreased at elevated $f$$CO_2$,
which is contrasting to laboratory studies showing that several species in this group benefit
from high $CO_2$ and can increase their growth rates (Tsuzuki et al. 1990; Yang & Gao 2003).

**4.2 Seston FAs**
The relative PUFA content of seston showed a significant decrease over time, which can be
attributed to nutrient depletion in the mesocosms, particularly silicate and phosphate
concentrations, which caused a decrease in dinophyta and heterokontophyta abundances. These
two groups of microalgae have been identified as a rich in PUFA content (Galloway & Winder
2015) and their decrease in the mesocosms explains the concomitant decrease in PUFA. Silicate
is required by heterokontophyta for the formation of new frustules during cell division, and



when limited, cell division stops (Flynn & Martin-Jézéquel 2000). Phosphorus is required for
the production of PUFA-rich membrane phospholipids during cell division and growth
(Guschina & Harwood 2009). Nutrient limitation, which causes reduced cell division rates,
results in a lower production of phospholipid and increased production of storage lipid,
primarily triacylglycerols (Guschina & Harwood 2009). Triacylglycerols tends to be rich in
SFA and MUFA; therefore the increase in triacylglycerols with nutrient limitation typically
resulted in decreased proportions of PUFA in most algae (Guschina & Harwood 2009). This is
consistent with our observations in the mesocosms, where the relative PUFA content of seston
followed the phosphate concentration. From this perspective one may expect that any $CO_2$
effect in algal PUFA will occur when cells are actively growing since nutrient limitation
(silicate and phosphorus) will have more profound consequences in the cell physiology than an
excess of $CO_2$.

The absence of a PUFA response to $CO_2$ differs with a report of an Arctic plankton community
showing an increase of PUFA at high $CO_2$ levels during part of a mesocosm experiment
experiencing nutrient additions (Leu et al. 2013). This was attributed to a change in the plankton
community composition due to a rise in abundance of dinoflagellates at high $CO_2$ (Leu et al.
2013). Our results show a decrease in PUFA due to a decline in dinoflagellates. The different
PUFA trend between these experiments can be attributed to the specific plankton community
composition and their related FA profiles alongside with phosphate and silicate limitation in
our study, which causes a reduction of the biomass of some PUFA-rich taxa. Species
composition of a natural plankton assemblage determines its food quality properties as distinct
algal taxonomic groups have different FA composition profiles (Galloway & Winder 2015). A
$CO_2$-driven change in the Arctic plankton community composition (Leu et al. 2013) promoted
the presence of species rich in PUFA. In our study the absence of a $CO_2$ response in taxa
composition and the apparent influence of phosphate and silicate limitation in the algal FA
composition resulted in a rather homogeneous PUFA concentration between $CO_2$ treatments.

**4.3 Copepod fatty acids**
Our results showed that the PUFA concentration of the dominating copepod species, *A. bifilosa*
and *E. affinis* did not vary between the different $CO_2$ treatments. However, the PUFA decrease
in both copepods over the experimental period reflects the decline in the PUFA content of the
seston. This observation is consistent with other studies showing that copepods strongly rely on



their diet as source of FA and that their composition, especially PUFA, mirrors the algae they
graze on (Ishida et al. 1998; Caramujo et al. 2007; Rossoll et al. 2012).

Several studies have shown a limited direct $CO_2$ effect in the copepods FA of some species,
like the genus *Acartia*, which is rather insensitive to projected high $CO_2$ exposure up to 5000
μatm $CO_2$ (Kurihara et al. 2004; Kurihara & Ishimatsu 2008). Copepods experience widely
varying pH conditions on a daily basis due during their vertical migration, shown in the same
area as the current study (Almén et al. 2014), which may explain their tolerance to pH
variations. Several studies have demonstrated that food quality of the available prey in terms of
PUFA content can affect egg production, hatching success and nauplii survival in copepods
(Jónasdóttir 1994; Caramujo et al. 2007; Jónasdóttir et al. 2009). Indirect adverse $CO_2$ effects
through the diet of primary consumers have been reported in laboratory and field experiments
(Rossoll et al. 2012; Locke & Sprules 2000). However, the absence of a $CO_2$-driven change in
the community composition of primary producers and the homogeneous algal FA composition
due to phosphate and silicate limitations masked any noticeable $CO_2$-related effects in the algae
FA profile that could have affected the copepods during our experiment.
**5 Conclusions**

Considering that the Baltic Sea is a coastal sea with a natural frequent and wide pH variability
(Omstedt et al. 2009), it can be expected that the effects of OA on plankton communities will
be rather small within the range of predicted values for this century (Havenhand 2012). A
reduced OA sensitivity in systems experiencing high $CO_2$ fluctuations is supported by our
results and other studies using communities from the Baltic (Thomsen et al. 2010; Nielsen et
al. 2010; Rossoll et al. 2013). However, in coastal upwelling areas undergoing an increase in
hypoxic events, it is likely that elevated $CO_2$ values as presently experienced by coastal
organisms and projected by the end of the century (Melzner et al. 2013) will be more recurrent
in the future (Feely et al. 2004), with the potential to affect various properties of plankton
communities.

Nonetheless, it is clear that the plankton community response to OA and concomitant effects
on its food quality for higher trophic levels will strongly depend on the sensitivity of primary
producers and on how OA affects the species composition of plankton assemblages (Leu et al.
2013; Rossoll et al. 2013). This result is important as any change in primary producers in terms



of FA, particularly essential biomolecules such as PUFA, may scale up in food webs since FAs
are incorporated into the lipids of larval fish (Fraser et al. 1989; Izquierdo et al. 2001).
Considering that fish is a critical natural resource (FAO, 2010), adverse OA effects on food
quality can reach up to human populations who rely on fisheries as an important food source
(Sargent et al. 1997; Arts et al. 2001).

## Acknowledgements

We thank the KOSMOS team and all of the participants in the mesocosm campaign for their
support during the experiment. In particular, we would like to thank Andrea Ludwig for co-
ordinating the campaign logistics and assistance with CTD operations and the diving team. We
also gratefully acknowledge the captain and crew of RV *ALKOR* (AL394 and AL397) for their
work transporting, deploying and recovering the mesocosms and the Tvärminne station and
staff for their logistic support. This collaborative project was funded by BMBF projects
BIOACID II (FKZ 03F06550) and SOPRAN Phase II (FKZ 03F0611).

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

**Figures and Figure Legends**


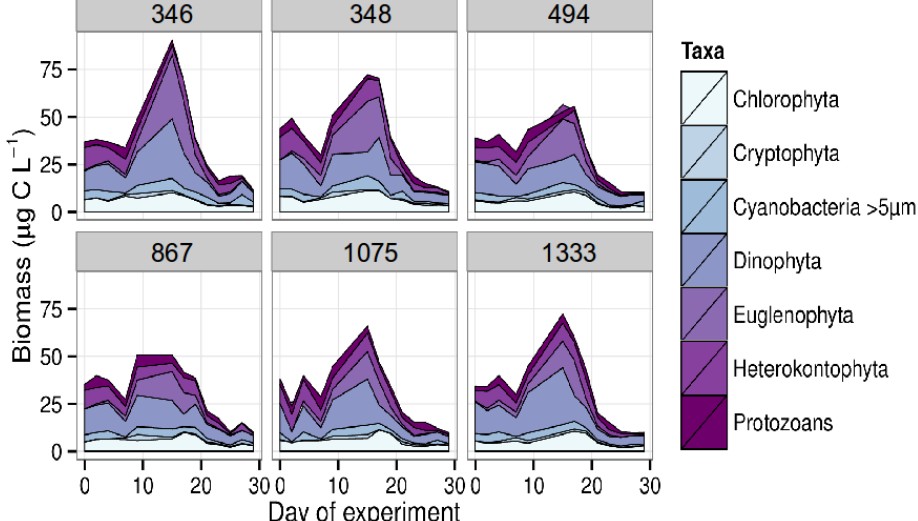


**Figure 1.** Calculated biomass after cell counts of the main plankton taxonomic groups in the

different $CO_2$ treatments between day 1 and 29. Each treatment is labeled with the average $f$CO$_2$

level of the entire experimental period (top).






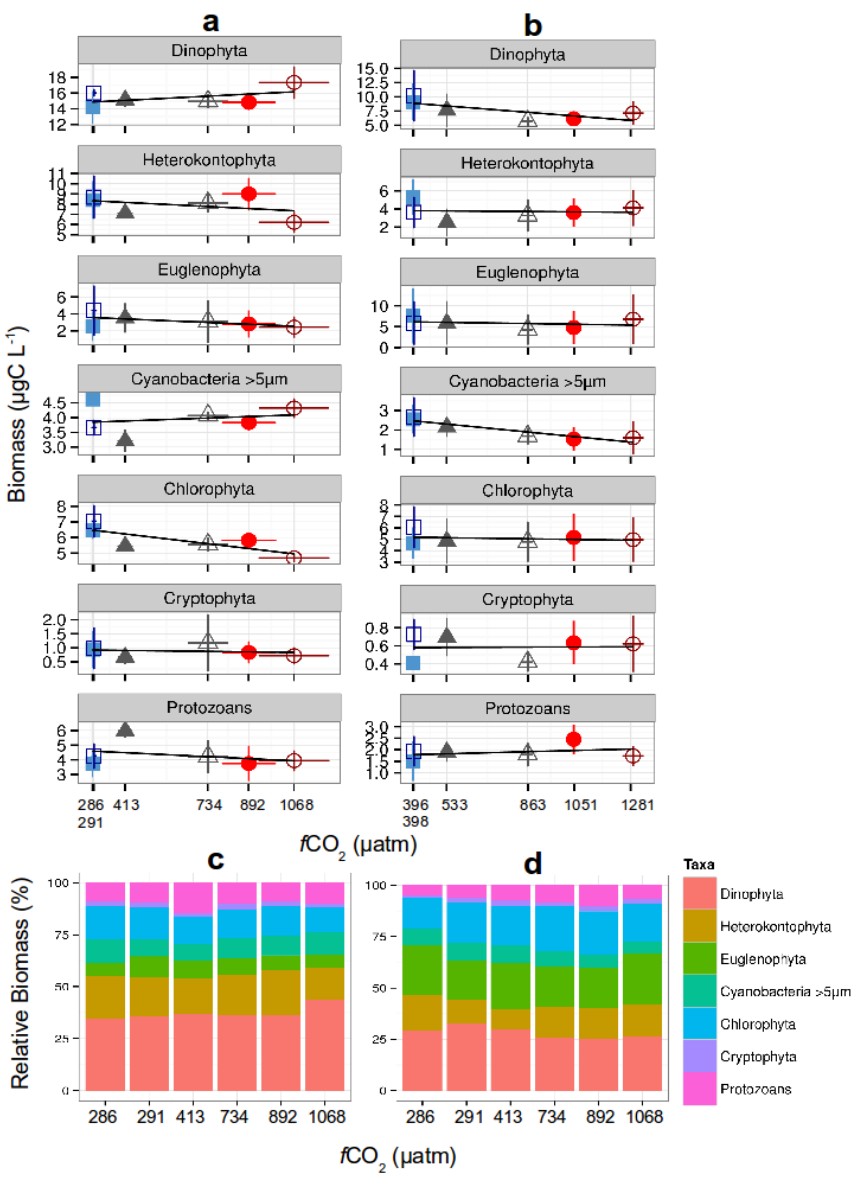

**Figure 2.** The top panels show the mean of the calculated biomass of each plankton taxon in a) Phase 1, between the days 0 to 15; and b) Phase 2, between days 15 to 29, in the $CO_2$ gradient treatments. The bottom panels show the relative biomass of the different plankton groups between c) Phases 1 and d) Phase 2. The x-axes show the measured average $fCO_2$ in each phase, error bars show standard error in a and b (n=5 for a; n=5 for b).

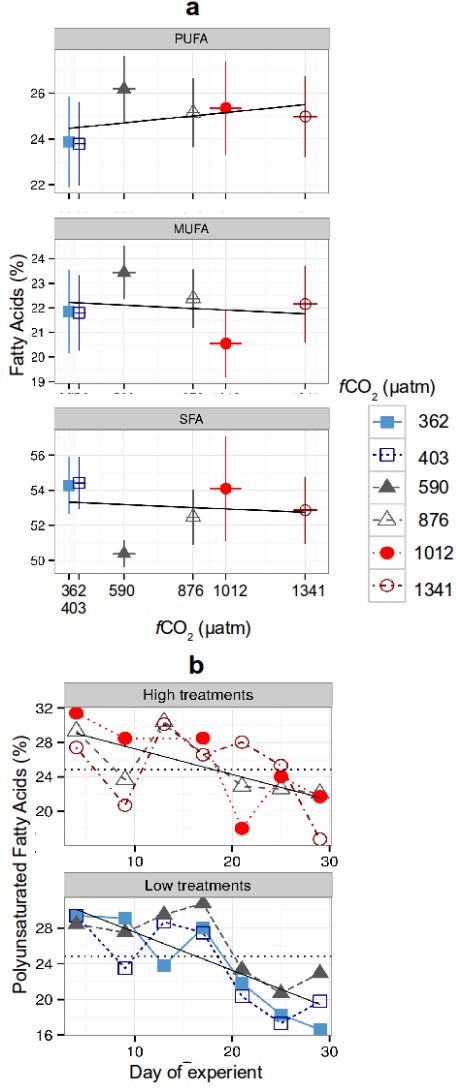

**Figure 3.** a) Relative polyunsaturated (PUFA), monounsaturated (MUFA), and saturated (SFA)
fatty acids content in the seston as a function of $f\mathrm{CO_2}$ between day 1 and 29. The x-axes show
the mean $f\mathrm{CO_2}$ measured during the sampling period, bars shows standard error. b) Relative
PUFA composition of the seston showed over time in the 876, 1012 and 1314 µatm $f\mathrm{CO_2}$ levels
(high $\mathrm{CO_2}$ treatments) and the 362, 403 and 590 µatm $f\mathrm{CO_2}$ levels (low $\mathrm{CO_2}$ treatments).
Horizontal dashed line indicates the position of the overall mean PUFA value.






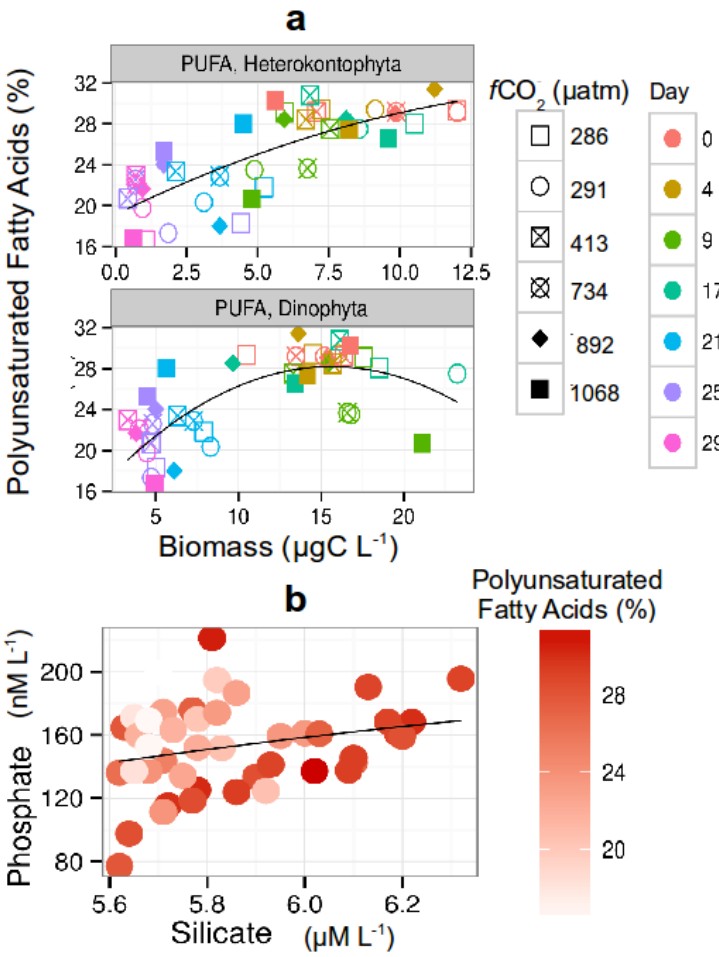

**Figure 4.** a) Relation between sestonic relative polyunsaturated fatty acids (PUFA) with
heterokontophyta (PUFA, heterokontophyta) and dinophyta (PUFA, dinophyta) biomass. b)
Relation between relative sestonic PUFA content with silicate and phosphate abundance in the
mesocosms.




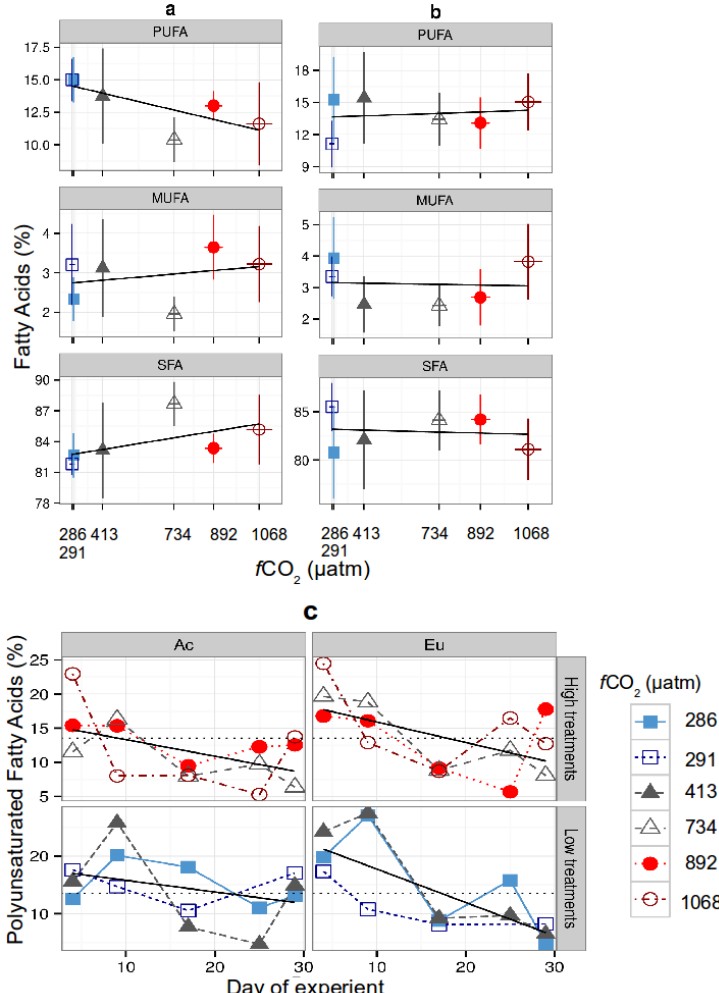


**Figure 5.** a) and b) show the relative polyunsaturated (PUFA), monounsaturated (MUFA), and saturated (SFA) fatty acids content in the copepods *Acartia bifilosa* and *Eurytemora affinis*, respectively, under the $f$CO$_2$ gradient treatments between day 1 to 29. The x-axes show the mean $f$CO$_2$ measured during the sampling period, bars shows standard error. c) Relative PUFA composition of *Acartia bifilosa* (Ac) and *Eurytemora affinis* (Eu) over time in the 876, 1012 and 1314 µatm $f$CO$_2$ levels (high CO$_2$ treatments) and the 362, 403 and 590 µatm $f$CO$_2$ levels (low CO$_2$ treatments). Horizontal dashed line indicates the position of the overall mean PUFA value.