# Peer review of "Effect of ocean acidification on the structure and fatty acid composition"

_Biogeosciences, 2015_

## Referee Comment (RC1) · Anonymous Referee #1 · 26 Jan 2016

General comments

The paper is devoted to very important problem: an influence of elevated carbon dioxide concentrations on aquatic trophic chains, namely on food quality for consumers, regarded as content of PUFA in microalgae. Thereby, the paper is of potential interest for pure and applied aquatic ecology. An effect of elevated $CO_2$ on PUFA content in some microalgae has been demonstrated previously for laboratory cultures, and it is worth to test it in mesocosm studies for natural phytoplankton communities. The mesocosm experiments were well designed and the $CO_2$ levels, predicted by some future scenario, were used. Fatty acids were measured both in phytoplankton (seston) and in dominant zooplankton species. However, fatty acids in the work were represented as

three groups only: SFA, MUFA and PUFA. To my mind, the group 'PUFA' is too coarse for the aim of study, namely for consideration of changing of food quality for consumers. Long-chain n-6 and n-3 PUFA act as physiological and biochemical counter-regulators in animals. To my mind, it is impossible to interpret a significance of their sum for animals' status. Physiological role of 18C PUFA for animals is unclear. As a matter of fact, EPA (20:5n-3) and DHA (22:6n-3) are the indicators of nutritive quality for zooplankton, rather than sum of unspecified PUFA, used in this study. Even if the sum of PUFA in seston (phytoplankton) stay the same during CO2 variations, it does not mean, that the nutritive quality for zooplankton also stays unchanged. For instance, decrease of 22:6n-3 vs. increase of 18:3n-3 in sum PUFA will decrease the nutritive quality for copepods. This can be due to a decrease of part of dynophytes vs. an increase of part of chlorophytes in phytoplankton (see Specific comments below). Hence, to my opinion, in the work an effect of the acidification on "fatty acid composition of a natural plankton community" was not studied, since fatty acid composition was not properly measured. Conclusion is irrelevant to results, obtained in the work, and resembles a mini-review of literature.

Specific comments

Page 5, line 149: Breteler et al. (1999) – should be Klein Breteler et al. (1999). Improve here and in Reference, line 347 - Klein Breteler W.C.M.

Page 5, line 167: (Breteler & Schouten, 1999) – is absent in Reference!

Pages 6-7, lines 207-211: the decrease of relative biomass of dynophyta and the concomitant increase of biomass of chlorophyta, to my mind, could result in decrease of content of 22:6n-3 and increase of 18:3n-3 in seston. However, using the coarse parameter 'PUFA' you had not an opportunity to see these changes in FA composition of zooplankton food, which likely significantly affected the nutritive quality of seston.

---

## Referee Comment (RC2) · Anonymous Referee #2 · 12 Feb 2016

The manuscript "Effect of ocean acidification on the structure and fatty acid composition of a natural plankton community in the Baltic Sea" by Bermúdez et al. is a well-written and interesting study on a topic of general interest. The aim of the study was to analyze the effect of increasing $CO_2$ concentrations and ocean acidification on a coastal plankton community of the eastern Baltic Sea. The authors hypothesize that coastal/brackish environments have a high tolerance to increased $CO_2$ concentrations, since they face naturally fluctuations in $CO_2$ and pH in coastal systems. The results support this hypothesis, since there was no significant difference in community composition, Seston and Copepod FA concentration between treatments with different $CO_2$ concentrations. FA composition changed over time in all $CO_2$ treatments, which was

explained by a change in community composition due to nutrient limitation.

Thus, the authors confirmed results found in the western Baltic Sea by Thomsen et al. (2010), Nielsen et al. (2010) and Rossoll et al. (2013), who also could not find any effect of elevated $CO_2$ concentrations on mussels and plankton communities. The introduction clearly explains the purpose of the study and presents it in a suitable context. Methods used in this study are appropriate and the results and figures well presented. The discussion addresses research questions posed in the introduction and interprets the results in light of previous knowledge. I think it would be very informative to include pH values of the different $CO_2$ treatments in the analysis, since pH is the actual factor that might affect marine organisms.

Otherwise, I have only few suggestions for improvement:

l. 50: "has increased the atmospheric concentration"

l. 122: Was pH measured in the $CO_2$ treatments?

l. 143: "algal fatty acid": better "seston fatty acid" as seawater samples include zooplankton. Were copepods removed before filtering seawater for seston fatty acid analysis?

l. 168: Is there information on individual fatty acids? Not all PUFAs are essential food for copepods.

l. 208/209: "decrease in dinophyta (...) (Fig. 2c)": Fig. 2c shows an increase in dinophyta? I guess the decrease from phase 1 (Fig. 2c) to phase 2 (Fig. 2d) is meant here?

l. 275: "have been identified as rich in PUFA"

l. 242: "MUFA and SFA increased in both species (Fig. S3)": Fig. S3 shows a decrease in MUFA and an increase in SFA in both copepod species.

l. 282: "Triacylglycerols tend to be rich"

l. 288: "consequences for the cell physiology"

l. 317: "daily basis during their vertical migration"

Figure 2a/b: legend for symbols is missing. However, I think here they are unnecessary since fCO2 is on the x-axis.

Figure 3a: same as Fig. 2a/b (and Fig. 5a/b). Why are fCO2 values different in each figure?

---

## Referee Comment (RC3) · Anonymous Referee #3 · 26 Feb 2016

**General comments**

I largely agree with reviewer 1 with respect to this manuscript's strengths and limitations: The study of OA effects on larger (mesocosm) scales to extend the knowledge gained from laboratory experiments is certainly required and the study of Bermúdez et al. is generally sound and the data analysed appropriately. On the other hand, the authors lose considerable amounts of information by pooling their FA results into the rather uninformative bulk categories SFA, MUFA and PUFA. Reporting details on particular, essential FA (such as EPA and DHA, see rev.1) may have yielded more insight into potential consequences of OA on phytoplankton and zooplankton lipid and community composition. Further, I do not understand why the authors report only relative

(%), rather than absolute FA amounts (e.g. $\mu$g FA / mg seston POC). This would have yielded important additional information on aspects of dietary quality of the phytoplankton community for the copepods and a discussion of potentially saturating or limiting quantities of essential PUFA. Summing up the panels of figure 1, it seems that the peak phytoplankton biomass shows a hyperbolic relationship with $CO_2$. This should be discussed. As the authors admit, the absence of strong OA effects on the FA composition of phyto- and zooplankton reported here is not particularly surprising for a low salinity and high variability system such as the Baltic Sea. Hence, although this study is conducted properly, it has limited appeal and a presumably low impact.

**Specific comments**

- L59 and 456: Do not cite unpublished work unless accepted for publication - L89-96: This whole paragraph is redundant with information stated previously. - L 144: Glass fibre filters do not have defined pore sizes - L151: The unit given for the IS addition ("ng/component $\mu$l") does not make sense - The panels of figure 2 a and b are vertically compressed with relatively large symbols and thus very hard to read. - The regressions in figure 3a should be plotted through the individual data points, rather than through the calculated mean values. I do not see the point of the PUFA figures 3b and 5c which could probably be removed.

Finally, the reference list is formatted sloppily. Some references are missing or misspelled, see rev.1. Number of listed/abbreviated authors, abbreviation of journal names, capitalization etc. vary a lot. Please revise carefully.

---

## Author Comment (AC1) · 13 May 2016

**Answer to comments on the manuscript bg-2015-669**

**Effect of ocean acidification on the structure and fatty acid composition of a natural plankton community in the Baltic Sea**

J.R. Bermúdez[1,2], M. Winder[3], A. Stuhr[1], A.K. Almén[4], J. E. Öst[4,5], U. Riebesell[1]

[1] {GEOMAR Helmholtz Centre for Ocean Research Kiel, Germany}

[2] {Facultad de Ingeniería Marítima, Ciencias Biológicas, Oceánicas y Recursos Naturales. Escuela Superior Politécnica del Litoral, ESPOL, Guayaquil, Ecuador}

[3] {Department of Ecology, Environment and Plant Sciences, Stockholm University, Stockholm, Sweden}

[4] {Novia University of Applied Sciences, Coastal Zone Research Team, Ekenäs, Finland}

[5] {Tvärminne Zoological Station, University of Helsinki, J.A. Palménin tie 260, FI-10900 Hanko, Finland}

Correspondence to: J.R. Bermúdez (jrbermud@espol.edu.ec)

**Reviewer #1**

**C:** Comment
**A:** Answer

**General comments**

**C:** The paper is devoted to very important problem: an influence of elevated carbon dioxide concentrations on aquatic trophic chains, namely on food quality for consumers, regarded as content of PUFA in microalgae. Thereby, the paper is of potential interest for pure and applied aquatic ecology. An effect of elevated $CO_2$ on PUFA content in some microalgae has been demonstrated previously for laboratory cultures, and it is worth to test it in mesocosm studies for natural phytoplankton communities. The mesocosm experiments were well designed and the $CO_2$ levels, predicted by some future scenario, were used. Fatty acids were measured both in phytoplankton (seston) and in dominant zooplankton species. However, fatty acids in the work were represented as three groups only: SFA, MUFA and PUFA. To my mind, the group 'PUFA' is too coarse for the aim of study, namely for consideration of changing of food quality for consumers.

Long-chain n-6 and n-3 PUFA act as physiological and biochemical counter-regulators in animals. To my mind, it is impossible to interpret a significance of their sum for animals' status. Physiological role of 18C PUFA for animals is unclear. As a matter of fact, EPA (20:5n-3) and DHA (22:6n-3) are the indicators of nutritive quality for zooplankton, rather than sum of unspecified PUFA, used in this study. Even if the sum of PUFA in seston (phytoplankton) stay the same during $CO_2$ variations, it does not mean, that the nutritive quality for zooplankton also stays unchanged. For instance, decrease of 22:6n-3 vs. increase of 18:3n-3 in sum PUFA will decrease the nutritive quality for copepods. This can be due to a decrease of part of dynophytes vs. an increase of part of chlorophytes in phytoplankton (see Specific comments below). Hence, to my opinion, in the work an effect of the acidification on "fatty acid composition of a natural plankton community" was not studied, since fatty acid composition was not properly measured. Conclusion is irrelevant to results, obtained in the work, and resembles a mini-review of literature.

**A:** We do agree with the reviewer. A complete set of figures of the different seston and zooplankton PUFA has bed added in the manuscript and is discussed in the main text. This data shows that only 18:2n6 and 18:2n3 have a positive significant correlation with $f$CO$_2$. Also the abundance of 18:2n6c, 18:3n3 is also affected by silicate, while 20:5n3c and 22:6n3c is actually driven by Silicate and Phosphate abundance.

**Specific comments**

**C:** Page 5, line 149: Breteler et al. (1999) – should be Klein Breteler et al. (1999). Improve here and in Reference, line 347 - Klein Breteler W.C.M.
**A:** Corrected.

**C:** Page 5, line 167: (Breteler & Schouten, 1999) – is absent in Reference!
**A:** Corrected. The reference does not belong there, was misplaced and therefore did not appear in the reference list.

**C:** Pages 6-7, lines 207-211: the decrease of relative biomass of dynophyta and the concomitant increase of biomass of chlorophyta, to my mind, could result in decrease of content of 22:6n-3 and increase of 18:3n-3 in seston. However, using the coarse parameter 'PUFA' you had not an opportunity to see these changes in FA composition of zooplankton food, which likely significantly affected the nutritive quality of seston.

**A:** This is possible since dinophyta are a better source of 22:6n-3 (Galloway & Winder 2015). However this was revised and there were no significant relations between the algae groups and this FA.

**Reviewer #2**

**General comments**

**C:** The manuscript "Effect of ocean acidification on the structure and fatty acid composition of a natural plankton community in the Baltic Sea" by Bermúdez et al. is a well-written and

interesting study on a topic of general interest. The aim of the study was to analyze the effect of increasing $CO_2$ concentrations and ocean acidification on a coastal plankton community of the eastern Baltic Sea. The authors hypothesize that coastal/brackish environments have a high tolerance to increased $CO_2$ concentrations, since they face naturally fluctuations in $CO_2$ and pH in coastal systems. The results support this hypothesis, since there was no significant difference in community composition, Seston and Copepod FA concentration between treatments with different $CO_2$ concentrations. FA composition changed over time in all $CO_2$ treatments, which was explained by a change in community composition due to nutrient limitation.

Thus, the authors confirmed results found in the western Baltic Sea by Thomsen et al. (2010), Nielsen et al. (2010) and Rossoll et al. (2013), who also could not find any effect of elevated $CO_2$ concentrations on mussels and plankton communities. The introduction clearly explains the purpose of the study and presents it in a suitable context.

Methods used in this study are appropriate and the results and figures well presented. The discussion addresses research questions posed in the introduction and interprets the results in light of previous knowledge. I think it would be very informative to include pH values of the different $CO_2$ treatments in the analysis, since pH is the actual factor that might affect marine organisms.

**A:** We are glad that the reviewer does find the article interesting.

The decision to do not add the pH data is to keep consistency with the other articles of the special issue where it is included. Doing the data analysis with pH instead of $fCO_2$ will naturally result in significant results where they are present. Also, adding more information may turn the manuscript a bit confusing without really improving it.

**Specific comments**

Otherwise, I have only few suggestions for improvement:

**C:** l. 50: "has increased the atmospheric concentration"
**A:** Corrected.

**C:** l. 122: Was pH measured in the CO2 treatments?
**A:** Yes, total pH was determined by spectrophotometry as described by Paul et al. (2015). This is now clarified in the manuscript.

**C:** l. 143: "algal fatty acid": better "seston fatty acid" as seawater samples include zooplankton. Were copepods removed before filtering seawater for seston fatty acid analysis?
**A:** Corrected. Yes, copepod were removed by filtering the water through a 100 μm net before sample collection.

**C:** l. 168: Is there information on individual fatty acids? Not all PUFAs are essential food for copepods.

**A:** Yes, this was a concern of the first and third review. This has been addressed (please see above).

**C:** l. 208/209: "decrease in dinophyta (: : :) (Fig. 2c)": Fig. 2c shows an increase in dinophyta? I guess the decrease from phase 1 (Fig. 2c) to phase 2 (Fig. 2d) is meant here?
**A:** Yes, that is what is mean; the paragraph has been reworded.

**C:** l. 275: "have been identified as rich in PUFA"
**A:** Corrected

**C:** l. 242: "MUFA and SFA increased in both species (Fig. S3)": Fig. S3 shows a decrease in MUFA and an increase in SFA in both copepod species.
**A:** We agree, this has been corrected.

**C:** l. 282: "Triacylglycerols tend to be rich"
**A:** Corrected.

**C:** l. 288: "consequences for the cell physiology"
**A:** Corrected.

**C:** l. 317: "daily basis during their vertical migration"
**A:** Corrected.

**C:** Figure 2a/b: legend for symbols is missing. However, I think here they are unnecessary since $f\text{CO}_2$ is on the x-axis.
**A:** Yes, the legend for symbols was not added since the $f\text{CO}_2$ values are given in the x-axis and thus avoid making the figure too busy.

**C:** Figure 3a: same as Fig. 2a/b (and Fig. 5a/b). Why are $f\text{CO}_2$ values different in each figure?
**A:** The same reason as in the previous comment.

The difference is due to the frequency of the sampling. Samples to determine $f\text{CO}_2$ were taken every day, while seston and zooplanktonic fatty acids (FA) were taken every fourth and seventh day, respectively (this is stated in material and methods). Since $f\text{CO}_2$ varies daily and the sampling days for seston and zooplankton FA were not always the same, the $f\text{CO}_2$ values for each set of samples is different.

**Reviewer #3**

**General comments**

**C:** I largely agree with reviewer 1 with respect to this manuscript's strengths and limitations: The study of OA effects on larger (mesocosm) scales to extend the knowledge gained from laboratory experiments is certainly required and the study of Bermúdez et al. is generally sound and the data analyzed appropriately. On the other hand, the authors lose considerable amounts of information by pooling their FA results into the rather uninformative bulk categories SFA,

MUFA and PUFA. Reporting details on particular, essential FA (such as EPA and DHA, see rev.1) may have yielded more insight into potential consequences of OA on phytoplankton and zooplankton lipid and community composition.

**A:** We do agree with the reviewer. As stated above in response to Reviewer 1, a complete set of figures of the seston and zooplankton PUFA has bed added in the manuscript and is discussed in the main text.

**C:** Further, I do not understand why the authors report only relative (%), rather than absolute FA amounts (e.g. _g FA / mg seston POC). This would have yielded important additional information on aspects of dietary quality of the phytoplankton community for the copepods and a discussion of potentially saturating or limiting quantities of essential PUFA.

**A:** We did not use absolute FA amounts because our data set did not allow us to do so, which is a pity as we consider that quantitative data is as important as qualitative information.

However, relative content is widely used in literature over absolute FA amounts. For instance Tsuzuki et al. (1990); Sato et al. (2003); Fiorini et al. (2010); Rossoll et al. (2012); Torstensson et al. (2013) and Wynn-Edwards et al. (2014) reported it in this form.

Also, relative content has less variability and uncertainty in relation to absolute FA amounts. When calculating absolute amounts, the inherent natural variability and error of two samples (FA and POC) is combined; even more so when samples are collected in "natural conditions", as the mesocosms, because the composition of the suspended material in each sample will not be the same.

**C:** Summing up the panels of figure 1, it seems that the peak phytoplankton biomass shows a hyperbolic relationship with $CO_2$. This should be discussed.

**A:** Although this is an important observation, such discussion is beyond the scope of the present manuscript. However this is discussed in detail by Spilling et al. (2016) which is part of the present issue; they show that community respiration rates were lower at high $CO_2$ levels, but did not detect any effect of increased $CO_2$ on primary production. The percent carbon derived from microscopy counts (both phyto- and zooplankton), of the measured total particular carbon (TPC) decreased from ~ 26 % at t0 to ~ 8 % at t31, probably driven by a shift towards smaller plankton (< 4 μm) not enumerated by microscopy.

**C:** As the authors admit, the absence of strong OA effects on the FA composition of phyto- and zooplankton reported here is not particularly surprising for a low salinity and high variability system such as the Baltic Sea. Hence, although this study is conducted properly, it has limited appeal and a presumably low impact.

**A:** We disagree in this regard. There is compelling evidence showing an important negative $CO_2$ effect in the food quality of primary producers (Locke & Sprules 2000; Rossoll et al. 2012; Torstensson et al. 2013; Bermúdez et al. 2015). As the reviewer mentions, the exact focus of the present article is to show that the previously observed negative $CO_2$ effects are not widespread,

and that natural environmental variability, with the concomitant adaptation to said variability, may hamper said effects.

**Specific comments**

**C:** - L59 and 456: Do not cite unpublished work unless accepted for publication.
**A:** Corrected.

**C:** - L89-96: This whole paragraph is redundant with information stated previously.
**A:** Corrected.

**C:** - L 144: Glass fiber filters do not have defined pore sizes
**A:** Corrected.

**C:** - L151: The unit given for the IS addition ("ng/component _l") does not make sense
**A:** Corrected.

**C:** - The panels of figure 2 a and b are vertically compressed with relatively large symbols and thus very hard to read.
**A:** Corrected.

**C:** - The regressions in figure 3a should be plotted through the individual data points, rather than through the calculated mean values.
**A:** The line is a trend line, not a regression line and its intention is to show if there is a tendency in the data, but is not part of a statistic analysis. The analysis of that data was done with a Mixed Effects ANOVA Model, which does not look for correlations. The use of the means in the plot is to reduce the noise and make the figures easier to understand.

**C:** I do not see the point of the PUFA figures 3b and 5c which could probably be removed.
**A**: Those figures are intended to show that there is a change in the seston PUFA which is followed by the zooplankton PUFA, and that said change is not related to $CO_2$ but is related to nutrients and time.

**C:** Finally, the reference list is formatted sloppily. Some references are missing or misspelled, see rev.1. Number of listed/abbreviated authors, abbreviation of journal names, capitalization etc. vary a lot. Please revise carefully.
**A**: Corrected

**References**

Bermúdez, R. et al., 2015. Long-Term Conditioning to Elevated $pCO_2$ and Warming Influences the Fatty and Amino Acid Composition of the Diatom *Cylindrotheca fusiformis*. *Plos One*, 10(5), p.e0123945.

Fiorini, S. et al., 2010. Coccolithophores lipid and carbon isotope composition and their variability related to changes in seawater carbonate chemistry. *Journal of Experimental Marine Biology*

*and Ecology*, 394(1-2), pp.74–85..

Galloway, A.W.E. & Winder, M., 2015. Partitioning the Relative Importance of Phylogeny and Environmental Conditions on Phytoplankton Fatty Acids. *Plos One*, 10(6), p.e0130053.

Locke, A. & Sprules, W.G., 2000. Effects of acidic pH and phytoplankton on survival and condition of Bosmina longirostris and Daphnia pulex. *Fisheries (Bethesda)*, pp.187–196.

Rossoll, D. et al., 2012. Ocean acidification-induced food quality deterioration constrains trophic transfer. *PloS one*, 7(4), p.e34737.

Sato, N., Tsuzuki, M. & Kawaguchi, A., 2003. Glycerolipid synthesis in Chlorella kessleri 11 h II . Effect of the CO2 concentration during growth. *Biochimica et Biophysica Acta*, 1633, pp.35 – 42.

Spilling, K. et al., 2016. Ocean acidification decreases plankton respiration: evidence from a mesocosm experiment. *Biogeosciences Discussions*, (January 2016), pp.1–35.

Torstensson, a. et al., 2013. Synergism between elevated $p$CO$_2$ and temperature on the Antarctic sea ice diatom *Nitzschia lecointei*. *Biogeosciences*, 10(10), pp.6391–6401.

Tsuzuki, M. et al., 1990. Effects of CO(2) Concentration during Growth on Fatty Acid Composition in Microalgae. *Plant physiology*, 93(3), pp.851–6.

Wynn-Edwards, C. et al., 2014. Species-Specific Variations in the Nutritional Quality of Southern Ocean Phytoplankton in Response to Elevated pCO2. *Water*, 6(6), pp.1840–1859.